# Obscurin Rho GEF domains are phosphorylated by MST-family kinases but do not exhibit nucleotide exchange factor activity towards Rho GTPases *in vitro*

Daniel Koch🄞¤*, Ay Lin Kho, Atsushi Fukuzawa, Alexander Alexandrovich, Kutti J. Vanaanen, Andrew Beavil🄞, Mark Pfuhl, Martin Rees🄞*, Mathias Gautel*

Randall Centre for Cell and Molecular Biophysics, King's College London, London, United Kingdom

¤ Current address: Max Planck Institute For Neurobiology of Behavior—Caesar, Bonn, Germany
* dkoch.research@protonmail.com (DK); martin.rees@kcl.ac.uk (MR); mathias.gautel@kcl.ac.uk (MG)

**Data Availability Statement:** All relevant data are within the paper and its Supporting Information files.

## Abstract

Obscurin is a giant muscle protein (>800 kDa) featuring multiple signalling domains, including an SH3-DH-PH domain triplet from the Trio-subfamily of guanosine nucleotide exchange factors (GEFs). While previous research suggests that these domains can activate the small GTPases RhoA and RhoQ in cells, *in vitro* characterization of these interactions using biophysical techniques has been hampered by the intrinsic instability of obscurin GEF domains. To study substrate specificity, mechanism and regulation of obscurin GEF function by individual domains, we successfully optimized recombinant production of obscurin GEF domains and found that MST-family kinases phosphorylate the obscurin DH domain at Thr5798. Despite extensive testing of multiple GEF domain fragments, we did not detect any nucleotide exchange activity *in vitro* against 9 representative small GTPases. Bioinformatic analyses show that obscurin differs from other Trio-subfamily GEFs in several important aspects. While further research is necessary to evaluate obscurin GEF activity *in vivo*, our results indicate that obscurin has atypical GEF domains that, if catalytically active at all, are subject to complex regulation.

## Introduction

Obscurin is a giant muscle protein of yet poorly understood functions [1, 2]. Its N-terminus links the proteins titin and myomesin at the sarcomeric M-band and the C-terminus of obscurin isoform A binds to small ankyrins at the sarcoplasmic reticulum membrane and at subsarcolemmal protein complexes, thereby creating multiple physical links that contribute to M-band stability and membrane integrity [3–9]. In addition to these structural roles, obscurin is likely involved in multiple signalling contexts as it features two kinase domains, an IQ-motif, two calmodulin binding regions and an SH3-DH-PH RhoGEF domain triplet [1, 10]. Small GTPases of the Rho-family are a class of important signalling proteins that cycle between a biologically inactive GDP-bound form and a functional GTP-bound form and regulate diverse cellular functions including gene expression, cytoskeletal remodelling and contractility

**Funding:** DK: grant [FS/17/65/33481], British Heart Foundation, https://www.bhf.org.uk/ MG: 201543/Z/16/Z, Wellcome Trust, https://wellcome.org/ Research grant I/81 797 and I/78 989, Volkswagen Foundation, https://www.volkswagenstiftung.de/en/foundation MG and MR: ERC-2019-SyG grant no. 856118, european research council, https://erc.europa.eu/ MG and ALK: MR/R003106/1, Medical research council, https://www.ukri.org/councils/mrc/ The funders had no role in study design, data collection and analysis, decision to publish, or preparation of the manuscript.

**Competing interests:** The authors have declared that no competing interests exist.

[11, 12]. Similar to phosphorylation being catalyzed by a kinase, transition to the active GTP state is promoted by guanosine nucleotide exchange factors (GEFs) [13]. Previous studies showed that the obscurin DH-PH domains may act as a GEF on GTPases RhoA and RhoQ/TC10 but not Rac1 or Cdc42 and that these interactions are important for myofibril organization and may alter gene expression in response to mechanical signals [14, 15]. However, these insights were obtained using co-IP techniques involving nucleotide-state modifying Rho mutants and effector-domain pulldown assays from cell homogenates. While these approaches are suitable for a first characterization, they offer little information on substrate specificity and kinetics and can be prone to unintended consequences resulting from GTPase-mutants, over-expression of the GEF and unidentified cellular factors [16, 17]. Characterising the regulation and mechanism of GEF activity itself is subject to similar experimental constraints. Moreover, obscurin has only been tested against four GTPases although the Rho-family features more than 20 members, many of which could be potential substrates of the obscurin RhoGEF domains [11, 18].

In contrast, biochemical, biophysical and structural *in vitro* methods allow for quantitative and precisely controlled characterization of kinetics, mechanism and regulation of GTPase: GEF interactions. Here, we report the production of soluble, stable recombinant obscurin GEF domains necessary for such studies. We identified kinases and phosphatases that can regulate the phosphorylation status of the obscurin RhoGEF region at a physiological site and tested over 10 different obscurin RhoGEF fragments against RhoA, RhoB, RhoC, Rac1, Rac2, Rac3, RhoG, Cdc42 and RhoQ/TC10 to determine kinetics, substrate specificity and regulation of the obscurin DH domain by adjacent domains and phosphorylation. Surprisingly, we found that none of the obscurin RhoGEF fragments exhibited any GEF activity. Bioinformatic analyses suggest that obscurin is an atypical member of the trio-subfamily of RhoGEFs to which it belongs and differs from other subfamily members at several important and conserved residues implicated in interactions with substrate GTPases, potentially explaining the lack of activity *in vitro*. We discuss implications for our understanding of obscurin RhoGEF function and outline avenues for future research.

## Material and methods

### Cloning and viral packaging

Human and chimeric obscurin GEF constructs 1–45 (cf. S2, S3 and S9 Figs) have been generated with standard PCR techniques using the primers listed in S1 Table followed by transfer of PCR products into a modified pET15b vector (Novagen) with an N-terminal hexahistidine-tag and a TEV cleavage site using the NEBuilder® HiFi DNA Assembly kit (NEB, cat. no. E2621S). C-terminally truncated (for removal of prenylation site) small GTPases RhoA$_{1\text{-}181, F25N}$, RhoB$_{1\text{-}188,F25N}$, RhoC$_{1\text{-}181,F25N}$, Rac1$_{1\text{-}177}$, Rac2$_{1\text{-}177}$, Rac3$_{1\text{-}177}$, RhoG$_{1\text{-}177}$, Cdc42$_{1\text{-}178}$ and RhoQ/TC10$_{1\text{-}185}$ were cloned into a modified pGEX-2TK vector (GE Healthcare) in which the thrombin cleave-site was substituted by a TEV cleavage site. Cloning of constructs 46–58 (cf. S10 Fig) and larg and vav2 DH domains was done by Bio Basic Inc (Canada, https://www.biobasic.com/). Cloning and viral packing of N-terminally eGFP-tagged human obscurin DH-PH (residues 5681–6019, numbering as in human obscurin B, NCBI ref. seq. NM_001098623) into either AAV9 or AdV vectors was performed by VectorBuilder Inc (United States, https://en.vectorbuilder.com/). All constructs were validated by sequencing.

### Protein expression and purification

For protein expression, BL21-CodonPlus (DE3)-RIPL (Agilent, cat. no. 230280) *E.coli* cells were heat-shock transformed with expression vectors and grown over night on LB agar plates

with carbenicillin. The next morning, 0.5 to 2L LB + carbenicillin were inoculated with cells scraped directly from the plate and grown at 37°C in a shaking incubator at 180–200 rpm until the culture reached an optical density at 600 nm (OD600nm) of 0.6–0.9, at which point expression was induced by addition of 0.2 mM IPTG and allowed to proceed overnight at 20°C. Cells were harvested the following morning and lysed via 2x freeze/thaw cycles followed by addition of lysozyme into the following lysis buffer: 50 mM Tris-HCl pH 7.5, 300 mM NaCl, 25 mM imidazole, 1:1000 (v/v) β-mercaptoethanol, (5mM $MgCl_2$ for GTPases), cOmplete™ EDTA-free Protease Inhibitor Cocktail (Roche, cat. no. 11873580001).

RhoGTPases were purified using a GSTrap™ 4B column (GE Healthcare, cat. no. 29048609) followed by size exclusion chromatography into a final buffer of 50 mM Tris-HCl, 100 mM NaCl, 2 mM $MgCl_2$ and 2 mM DTT. Obscurin SH3, SH3-DH, $DH_{5681-5889}$ and $DH_{5667-5899}$ fragments were purified using a HisTrap™ FF column (GE Healthcare, cat. no. 29048609) followed by size exclusion chromatography into a final buffer of 30 mM Hepes pH 7.5, 100 mM NaCl and 2 mM DTT. Chimeric and zebrafish obscurin fragments, vav2 and larg DH domains were batch-purified using a single step affinity enrichment with Ni-NTA resin (Qiagen, cat. no. 30410) followed by 3x washing in lysis buffer before proteins were eluted from the resin and buffer exchanged into final buffer (30 mM Hepes pH 7.5, 100 mM NaCl and 2 mM DTT). Following purification, each protein was aliquoted, flash-frozen in liquid $N_2$ and stored at -70°C until further use.

For small scale test expression and purification for the purpose of solubility screening, cells were grown for 3–4 days in 4 mL autoinduction medium [19] supplemented with carbenicillin in 24 deep-well plates (Elkay, cat. no. 43001–0066). Following transformation, cells were grown for ~6hrs at 37°C before the temperature was lowered to 20°C and cells were grown until harvesting. Expression of proteins was monitored using cells transformed with GFP or mCherry as indicator cultures. After pelleting, cells were lysed in the plates using Bacterial Protein Extraction Reagent (Thermo Scientific, cat. no. 78243) and purified with 200 μl of Ni-NTA resin (Qiagen, cat. no. 30410). After two washing steps (buffer A: 50mM Tris-HCl pH 7.7, 300 mM NaCl, 2 mM DTT, 30 mM imidazole), the protein was eluted with a volume of 2x400 μL elution buffer (like buffer A but with 1M imidazole).

Protein concentration was determined either by absorbance at a wavelength of 280 nm using a DS-11 FX+ Spectrophoto-/Fluorometer (DeNovix) or for GTPases, due to interference of the guanosine nucleotide, by the Bradford assay (VWR Life Sciences, cat. no. M172-1L) using BSA as a standard.

## Dot-blotting

For analysis of protein solubility, 1–2 μl samples from the purification screen were applied to a nitrocellulose membrane and allowed to dry for 1–2 hrs before the membrane was blocked for 15–30 min in with 5% (w/v) milk powder in antibody binding buffer (10 mM Tris pH 7.4, 9 g/l NaCl, 1 ‰ (v/v) Tween-20). Mouse anti-His mAb (Millipore, cat. no. 70796) was applied at a dilution of 1:1000 for 1hr at RT or overnight at 4°C. Secondary antibodies (HRP-conjugated rabbit anti-mouse IgG, Dako, cat. no. P0260) were applied at a dilution of 1:1000 for 1 h at room temperature before signals were detected by chemiluminescence on a ChemiDocTM XRS+ imaging system (Bio-Rad) using Clarity ECL Western Substrate (Bio-Rad, cat. no. 1705061).

## Buffer exchange

To exchange buffer of proteins, the following desalting columns appropriate for the sample volume were used according to manufacturer's instructions: 30–130 μL sample volume with

Zeba™ Spin Desalting Columns, 7K MWCO (Thermo Scientific™, cat. no. 89882) or illustra™ NAP-5/10/25 columns for sample volumes of 0.5/1/2.5 mL (GE Healthcare, cat. nos. 17-0853-01/17-0854-01/17-0852-01).

## 1D-NMR

We used 1D-NMR to analyse selected protein fragments and assess whether the resulting NMR spectra exhibit features that indicate correct folding of the domain [20]. The protein to be analysed was buffer exchanged into 25 mM $HNa_2PO_4$ pH 7.3, 100 mM NaCl, 2 mM DTT. The protein was concentrated to at least 60 μM and spectra were recorded with an Ascend 600 instrument (Bruker Scientific Instruments).

## Nucleotide exchange kinetics

Nucleotide exchange kinetics of RhoGTPases were measured as described previously [21, 22]. Briefly, GTPases were preparatively pre-loaded with fluorescent mant-GDP by adjusting the protein concentration to 50–100 μM and spiking the sample with a 5-fold molar excess of mant-GDP (Jena Bioscience, cat. no. NU-204S) and 5 mM EDTA. After incubation of 2–4 hrs at RT in a light-protected tube, the buffer was exchanged to 50 mM Tris-HCl, 100 mM NaCl, 2 mM $MgCl_2$ and 2 mM DTT and the protein was aliquoted, flash-frozen in liquid $N_2$ and stored at -70˚C until further use.

Nucleotide exchange reactions were recorded on a CLARIOstar microplate reader (BMG LABTECH, Germany) in 96-well microplates (greiner bio-one, cat. no. 675076 or invitrogen, cat. no. M33089) at 25˚C. At the beginning of each experiment, 80 μL of 1.875 μM [GTPase] in reaction buffer equilibrated to room temperature (40 mM Hepes pH 7.5, 100 mM NaCl, 5 mM $MgCl_2$, 2 mM DTT) were pipetted into the plate and the signal was recorded (excitation wavelength $\lambda_{ex}$ = 360 nm, measured emission wavelength $\lambda_{ex}$ = 440 nm, sampling every 2-15s) for a sufficiently long period to reach a stable baseline (about 300 s). The nucleotide exchange reaction was initiated by addition of 20 μL of GDP or GDP + EDTA or GDP + GEF in buffer so that the final concentrations were 1.5 μM for the GTPase, 200 μM GDP, 10mM EDTA, or 0.5–25 μM GEF. After mixing by careful pipetting, the signal was recorded for further 1–2 hrs. The obtained traces were normalised by the average fluorescence signal of the first 300 s before start of the reaction and apparent rate constants $k_{obs}$ were obtained from the normalised traces by fitting the decay phase of the signal to a mono-exponential decay function of the form $f(t) = (a_0 - a_{plateau})\, e^{-k_{obs} \times t} + a_{plateau}$ in the GraphPad Prism software v8.3 (GraphPad Software, Inc).

## Kinase screening

Kinase screening using the obscurin SH3-DH domain as a substrate was performed externally with the KinaseFinder screen from ProQinase GmbH (now Reaction Biology Europe GmbH, https://www.reactionbiology.com/). Details on the procedure provided by ProQinase can be found in S1 Text.

## Phosphorylation of obscurin

For analytical purposes, 0.2 mg/ml obscurin RhoGEF fragments (5–20 μM, depending on the fragment used) were phosphorylated by 200 U PKA (Sigma-Aldrich, cat. no. 539576) or 500 U CaMK II (NEB, cat. no. P6060L) or 25 ng CaMK Id (BioVision, cat. no. 7713–5) or 25 ng MST2 (Millipore, cat. no. 14–524) in kinase reaction buffer (30 mM Hepes pH 7.5, 100 mM NaCl, 2 mM $MgCl_2$, 300 μM ATP, 2 mM DTT [+1mM $CaCl_2$ and 1 μg/ml (approx. 3 μM)

Calmodulin (NEB, cat. no. 6060S) for CaMK II/Id]). Samples were incubated for 3 h at 30°C in a PCR cycler and the reaction was stopped by addition of SDS sample buffer. For visualisation of phosphoprotein, 2 μg of obscurin substrate were separated by SDS-PAGE and stained with Pro-Q$^{TM}$ Diamond Phosphoprotein Gel Stain as described below. For preparative MST2-phosphorylation of obscurin SH3-DH or DH$_{5667-5899}$, 500 ng MST2 were added to a reaction volume of 200 μL kinase reaction buffer containing a substrate concentration of 110 μM and the reaction was allowed to proceed for 5 hrs at 25°C, followed by further incubation overnight at 8°C. The phosphorylated protein was buffer exchanged into 40 mM Hepes pH 7.5, 100 mM NaCl and 2 mM DTT.

## Dephosphorylation of MST2-phosphorylated obscurin

Following preparative phosphorylation of obscurin SH3-DH by MST2, 0.2 mg/ml obscurin phosphosubstrate were incubated with 1 μM of phosphatases PP1 or PP2A (Cayman Chemical, cat. no. 10011237) or 1 U rAPid Alkaline Phosphatase (Roche, cat. no. 4898133001) in 40 mM Hepes pH 7.5, 100 mM NaCl, 5mM MgCl$_2$ and 2 mM DTT (+ 1mM MnCl$_2$ for activation of PP1) in a reaction volume of 50 μL. The reaction was allowed to proceed for 1.5 h at 25°C and was stopped by the addition of SDS sample buffer.

## Phosphoprotein staining

For detection of protein phosphorylation on polyacrylamide gels, Pro-Q$^{TM}$ Diamond Phosphoprotein Gel Stain (Invitrogen, cat. no. P33301) was used according to the manufacturer's instructions.

## Mass-spectrometry

The MST2 phosphorylation site in the SH3-DH fragment was identified externally using a commercial mass-spectometry service from the Metabolomics and Proteomics Laboratory of the Bioscience Technology Facility of the University of York. Details on the procedure can be found in S2 Text.

## Neonatal ventricular rat cardiomyocyte preparation

Neonatal ventricular rat cardiomyocytes (NRCs) were isolated from Wistar rat pups and cultured as described previously [23]. In brief, hearts were isolated from Wistar rat pups at postnatal day 0 to 2 and cut into 4 in ice cold ADS (116 mM NaCl, 20 mM HEPES, 0.8 mM NaH2PO$_4$, 5.6 mM glucose, 5.4 mM KCl, 0.8mM MgSO$_4$ pH 7.35). The hearts were enzymatically digested in a sequential manner by incubation in enzyme solution containing collagenase type II (Worthington) (57.5 U/ml) and pancreatin (Sigma) (1.5 mg/ml) for 4–5 times for 15 min in a shaking incubator at 37°C. The supernatant is collected into medium containing 5% FCS and passed through a 70 micron cell strainer (Falcon Corning) before being pelleted at low speed. The cells were preplated onto 90 mm dishes (Nunc) in plating medium (DMEM, 5% FCS, 10% HS, non-essential amino acids, penicillin/streptomycin (P/S) and L-glutamine) for 2 h to allow non-myocytes to adhere. The non-adherent cardiomyocyte enriched fraction is then plated onto collagen (Attachin, Genlantis) coated 35mm dishes (Nunc) and cultured at 37°C and 5% CO$_2$. Once the cells had recovered (2–3 days), the non-adherent cells were washed away with culture medium (M199, DBSSK [116mM NaCl, 1 mM NaH2PO$_4$, 0.8 mM MgSO$_4$, 32.1 mM NaHCO$_3$, 5.5 mM glucose, 1.8 mM CaCl$_2$ pH7.2], 4% Horse serum, P/S and L-glutamine) and cultured until day 8–9 for further maturation (medium exchange every 3 days). The NRCs were transduced with 1μl of AdGFP or AdDHPH (at 1.05 x 10$^{11}$ and 9.16 x 10$^{10}$ IFU/ml respectively) for 24 hours then cultured in the

absence and presence of phenylephrine (100µM, Sigma) for another 24hours before being fixed with 4% paraformaldehyde for 10 mins.

## Immunostaining

The cells were permeabilised with 50µg/ml digitonin and blocked with 10% goat serum in IF buffer (10% bovine serum albumin, 0.1mM Tris pH7.5, 15.5mM NaCl, 0.2mM EGTA, 0.2mM MgCl$_2$) for 30mins at room temperature before incubation with primary antibody (rabbit anti-rhoQ/TC10 Abcam, cat. no. ab32079, mouse anti-rhoA Sigma, cat. no. SAB140017, both 1:100, rabbit anti titin Z1Z2 1:100 [24], mouse anti-myomesin b4 1:50 [25]) overnight at 4˚C. The cells were then washed 3 times 5 mins in PBS, and incubated with secondary antibody (Cy3 anti mouse, JIR, cat. no. 115-165-146, Cy5 anti rabbit, JIR, cat. no. 111175144 111561, DAPI, Sigma, all 1:100) for 1 hr at room temperature. After washing 3x 5 mins with PBS, the dishes were mounted with mounting medium (30mM Tris pH 9.5, 0.24M n-propyl gallate, 70% glycerol) and a coverslip applied and sealed with clear nail varnish before being imaged on a Zeiss LSM 510 confocal microscope.

## Western blotting

HEK293 cells were transduced with 1µl of AdGFP or AdDHPH (at 1.05 x 10$^{11}$ and 9.16 x 10$^{10}$ IFU/ml respectively) for 24 hours. The cells were scraped off the dish and resuspended in SDS-loading buffer (Laemmli). Homogenate samples or recombinant GTPases were loaded onto a 4–15% acrylamide gel (Biorad). The proteins were transferred onto nitrocellulose overnight at 60mA and blocked with 10% nonfat dry milk in lo-salt buffer (0.9% w/v NaCl, 10mM Tris pH 7.4, 0.1% tween-20) and probed with anti-HIS (Millipore, cat. no. 70796), anti-GFP (Roche) or anti-obscurin DH antibody [1] for 1 hour at RT. The blot was washed in lo-salt buffer 3 x 5 mins and incubated with HRP-tagged secondary antibodies (DAKO) for 1 hour. The blot was then washed again and signals were detected by chemiluminescence on a ChemiDocTM XRS + imaging system (Bio-Rad) using Clarity ECL Western Substrate (Bio-Rad, cat. no. 1705061).

## Bioinformatic analyses

Multiple sequence alignment of amino acid sequences was performed with clustal Omega server v1.2.4. (EMBLEBI, https://www.ebi.ac.uk/Tools/msa/clustalo/) [26]. Aligned sequences and amino acid properties such as hydrophobicity or percent identity across aligned sequences) were visualised in the UGENE software v1.28.1 (Unipro) [27].

Sequence conservation across species species was assessed with ConSurf (https://consurf.tau.ac.il/) [28] with the following parameters: amino acids, no known structures, no MSA upload (sequence was provided in FASTA format), Proteins database: Uniprot, Select homologs for ConSurf analyses: automatically. Other parameters were left at their default settings.

Secondary structure predictions for proteins were obtained from 2018 to 2019 using the PredictProtein (https://predictprotein.org/) and PSIPRED 4.0 (http://bioinf.cs.ucl.ac.uk/psipred/) with default parameters [29, 30].

Homology based tertiary structure models were obtained with I-TASSER (https://zhanggroup.org/I-TASSER/) at default parameters [31] and visualised with PyMOL Molecular Graphics System v2.3.4 (https://pymol.org/2/, Schrödinger, LLC).

## Molecular dynamics simulations

Before the simulations, a predicted structure of the obscurin SH3-DH-PH domain triplet was generated using the ColabFold AlphaFold2 server [32, 33] using default options with Amber

relaxation and a protein sequence corresponding to residues 5602 to 6008 of obscurin transcript variant 1. Prior to molecular dynamic simulations, phosphorylated models were generated in Coot [34] by either replacement of Ser5669 with phosphoserine, replacement of Thr5798 with phosphothreonine or replacement of both.

All-atom, solvated molecular dynamics simulations were run following energy minimisation for the four SH3-DH-PH models in Amber [35] using the ff14SB forcefield with a sampling rate of 100ps, temperature of 298k, ionic strength of 0.1M and a simulation time of 1us. The simulations were repeated 20 times for each model. Data including RMSD and RMSF was extracted from the molecular dynamics trajectories using scripts from ccptraj [36] and Bio3D [37], and the means of RMSD values were assessed for statistical differences using ANOVA in GraphPad Prism version 9.5.0.

Principal component (PC) analysis was run using the MDTraj and SciPy modules [38], using every 5th frame from the merged trajectories. 20 PCs were produced, which explained ~60% of the total structural variance, with the first 3 PCs individually explaining over 5% of the variance. PC heatmaps were plotted using the seaborn module and representative structures of PC minima and maxima were written out using the MDTraj module [39, 40].

The SH3-DH-PH structure generated every 5th frame of the four trajectories was aligned to the DH-PH domains of Dbs in the crystal structure of Dbs in complex with RhoA (PDB ID: 1LB1) [41] and the distance between SH3-DH-PH and RhoA atoms in the "clashing region" (residues 62–78 for Obscurin and 120 to 133 for RhoA) were measured to identify clashing atoms using the MDanalysis module distance_array function [42, 43].

## Results

### Construct design and purification of human obscurin RhoGEF domains

Initial attempts to purify recombinant, His$_6$-tagged obscurin RhoGEF domains (SH3-DH-PH, DH-PH, DH and PH) following expression in *E. coli* were not successful in our hands. Domain boundaries of these constructs were based on exon boundaries. We were able, however, to successfully express and purify obscurin SH3 and SH3-DH at high purity and at a yield of >5–10 mg/L culture (**S1 Fig**). Since these protein fragments were stable at high concentrations without precipitation, we reasoned that the insolubility of the catalytic DH domain may be a result of suboptimal domain boundaries. We thus performed a bioinformatic analysis of the amino acid sequence of obscurin RhoGEF domains using multiple sequence and structure prediction algorithms and found that strict adherence to exon boundaries may disrupt the first N-terminal alpha-helix of the obscurin DH domain and excludes several highly conserved residues (**Fig 1A**). Using this information, we designed and screened 39 additional constructs of the DH, PH and DH-PH domains and found that DH domain constructs that fully include the predicted N-terminal extension resulted in soluble protein fragments at the correct molecular weight in small-scale purification assays (**S2–S4 Figs**). DH-PH or PH domain constructs, however, did not result in soluble protein, despite different domain boundaries, indicating that the PH domain is intrinsically unstable and difficult to purify. Larger scale purification of two selected DH domain fragments (the larger encompassing adjacent linker regions and the smaller comprising only the DH domain) resulted in milligrams of highly pure, stable and folded protein (**Fig 1B–1D** and **S5 Fig**).

### Human obscurin RhoGEF domains do not exhibit catalytic activity towards Rho GTPases in vitro

The availability of highly pure and folded protein domains prompted us to characterize the catalytic activity of the obscurin RhoGEF domains towards Rho GTPases using enzyme kinetic

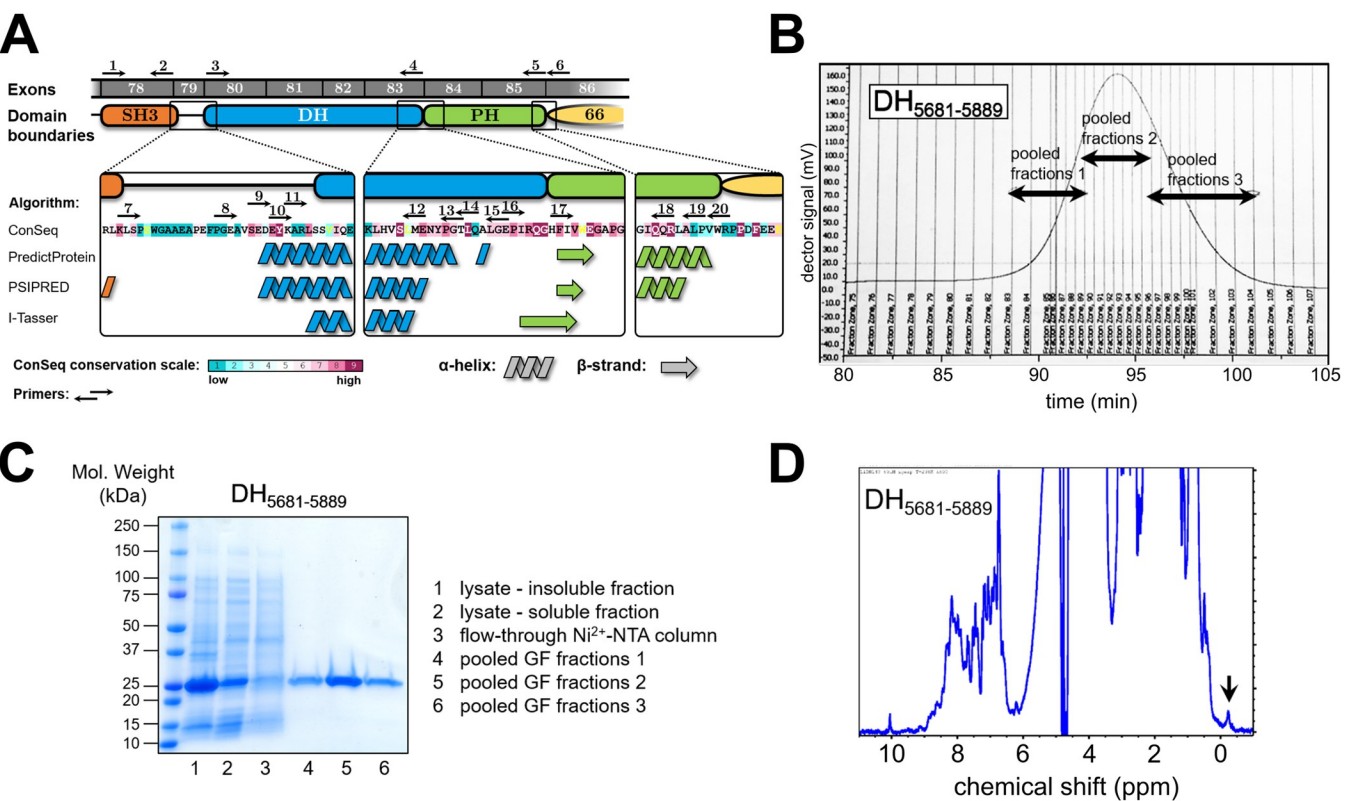

**Fig 1. Design and purification of obscurin RhoGEF fragments.** A, obscurin RhoGEF region. Exon boundaries and bioinformatic analysis and structure prediction based on amino acid sequence of human obscurin B isoform (NCBI ref. seq. NM_001098623). B, gel-filtration elution profile of DH domain fragment comprising residues 5681–5889. C, SDS-PAGE analysis of samples at different steps of the purification process of the DH$_{5681-5889}$ fragment. D, 1D-NMR analysis of purified DH$_{5681-5889}$ fragment. NMR spectrum exhibits wide peak dispersal and peaks below 0 ppm (black arrow), indicating that the protein is folded.

methods. Using GTPases prepared with the fluorescent nucleotide 2'/3'-O-(N-Methyl-anthraniloyl)-GDP as a substrate, we tested whether obscurin SH3-DH and DH fragments can facilitate nucleotide exchange on any of the typical Rho GTPases that cycle between GDP/GTP states and are thus regulated by GEFs (with the exception of RhoJ, which was not soluble in our hands). Surprisingly, we found that the nucleotide exchange rate after addition of obscurin RhoGEF fragments was indistinguishable from the intrinsic nucleotide exchange rate–in stark contrast to the positive controls (**Fig 2** and **S6 Fig**). In other words, neither of the obscurin RhoGEF fragments exhibited nucleotide exchange activity towards any of the tested Rho GTPases, including RhoA and RhoQ/TC10.

To confirm these results, we performed different pull-down experiments using a previously described protocol to stabilize the nucleotide-free high-affinity GEF-GTPase complex to specifically enrich GEFs [22]. Using the putative substrate RhoA as a bait, we found that regardless of whether RhoA is bound to nucleotides, the bare DH domain does not bind to RhoA, suggesting RhoA is neither a substrate, nor binding partner of obscurin's DH domain *in vitro* (**S7 Fig**). Our findings thus disagree with previous reports of obscurin RhoGEF domains activating RhoA and RhoQ/TC10 [14, 15].

To understand the reason for the discrepancy posed by our results, we further analyzed the sequence and predicted structure of obscurin RhoGEF domains, allowing us to better interpret our results in the context of the currently available mechanistic paradigms for Rho GEFs. The general mechanisms of DH-domain GEFs are well characterized and reviewed in the literature

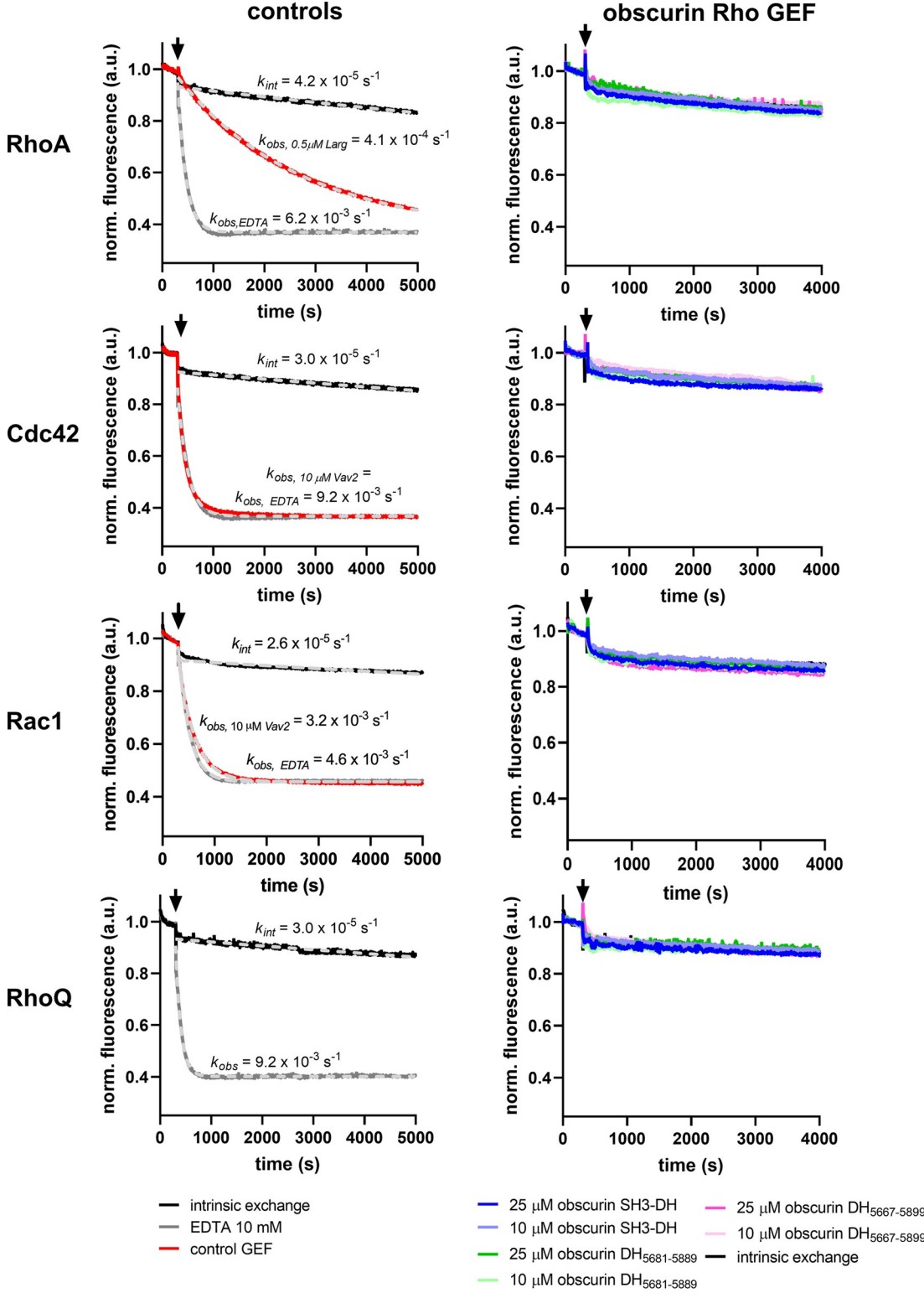

**Fig 2. Guanosine nucleotide exchange factor activity of obscurin RhoGEF fragments towards RhoA, Cdc42, Rac1 and RhoQ/ TC10.** Left panels show the intrinsic nucleotide activity as a negative control, EDTA and GEFs Vav2 and Larg as positive controls for each GTPase. Right panels show the nucleotide exchange activity after addition of obscurin. Black arrows indicate addition of buffer/GEF/EDTA. Data represent mean of n = 2–3 experiments.

[13, 41, 44, 45]. Due to the amino acid sequence of its RhoGEF domains, obscurin belongs to the Trio-subfamily of RhoGEFs, which includes the GEFs dbl and dbs, for which detailed structural and kinetic data are available [46]. Interestingly, the DH domains of dbl, dbs and trio have a much lower nucleotide exchange factor activity in the absence of their corresponding PH domain [47].

Subsequent work demonstrated that residues at the DH-PH domain interface (including residues from the PH domain) make contact with the switch II region of the substrate GTPase and contribute to the catalytic activity of the GEF (**Fig 3A and 3B**) [41, 45]. These residues are also present in most other trio-subfamily GEFs and are conserved in the obscurin sequence in humans, mice, chicken and zebrafish, but not in drosophila or nematodes (**S8 Fig**) [45].

This suggests the catalytic contribution of PH-domain residues is evolutionary conserved both in the trio-subfamily and across species. We thus hypothesized that these residues may need to be present for obscurin to exert GEF activity towards Rho GTPases. Since we were unable to purify any proteins containing the human obscurin PH domain, we cloned and attempted to purify several chimeric obscurin DH-PH fragments containing the human DH domain followed by the PH domain from human dbs and trio and the obscurin PH domain from chicken or zebrafish, all of which feature the aforementioned residues involved in supporting GEF activity. Of these chimeric proteins, only human/zebrafish chimeras (henceforth called "mermaid" obscurin) could be successfully purified (**S9 Fig**). Many zebrafish RhoGT-Pases (in particular the putative substrate RhoA) have >90% sequence identity with human RhoGTPases [48]. Given the high sequence identity of the substrates across species, we reasoned that zebrafish obscurin GEF fragments should likely exhibit activity towards human RhoGTPases.

We therefore cloned and successfully purified additional mermaid as well as several zebrafish obscurin DH and DH-PH fragments (**S10 Fig**). Next, we tested mermaid and zebrafish RhoGEF fragments for nucleotide exchange activity towards RhoA, Cdc42, Rac1 and RhoQ/TC10. Again, we found no increase in the nucleotide exchange rate after addition of these GEF domains (**Fig 3D**). We also tested the nucleotide exchange activity of zebrafish obscurin DH-PH$_{5884-6217}$ towards RhoB, RhoC, Rac2, Rac3 and RhoG but did not detect any increase in the nucleotide exchange rate (**S11 Fig**).

Our data show, therefore, that the inclusion of the PH domain was not sufficient to obtain catalytically active obscurin RhoGEF *in vitro*, at least for the tested substrates.

## Obscurin RhoGEF domains can be phosphorylated by MST kinases and CaMKs

The complete absence of guanosine nucleotide exchange activity of DH-PH fragments is peculiar. Flanking SH3 and PH domains as well as linker regions are known to sometimes inhibit the activity of the DH domain [49–53]. Typically, release of such autoinhibitory regions is mediated by phosphorylation (reviewed e.g. in [12, 54]). In the case of obscurin, simple autoinhibition of the obscurin DH domain by the SH3 or PH domain as the sole explanation for the lack of obscurin GEF activity is unlikely given the absence of activity of the isolated DH domain alone and the conserved positive catalytic contribution of the PH domain in trio-subfamily RhoGEFs. Interestingly, however, the murine obscurin RhoGEF region, too, has been reported to become phosphorylated at the residues corresponding to human Ser5669 (in the SH3-DH interdomain linker) and Thr5798 (within the DH domain) upon muscle exercise [55]. Ser5669 is conserved among human, mice and zebrafish, Thr5798 only among human and mice (**S12 Fig**). We thus speculated that phosphorylation might still be important to activate obscurin RhoGEF function, possibly via conformational changes in one of its domains.

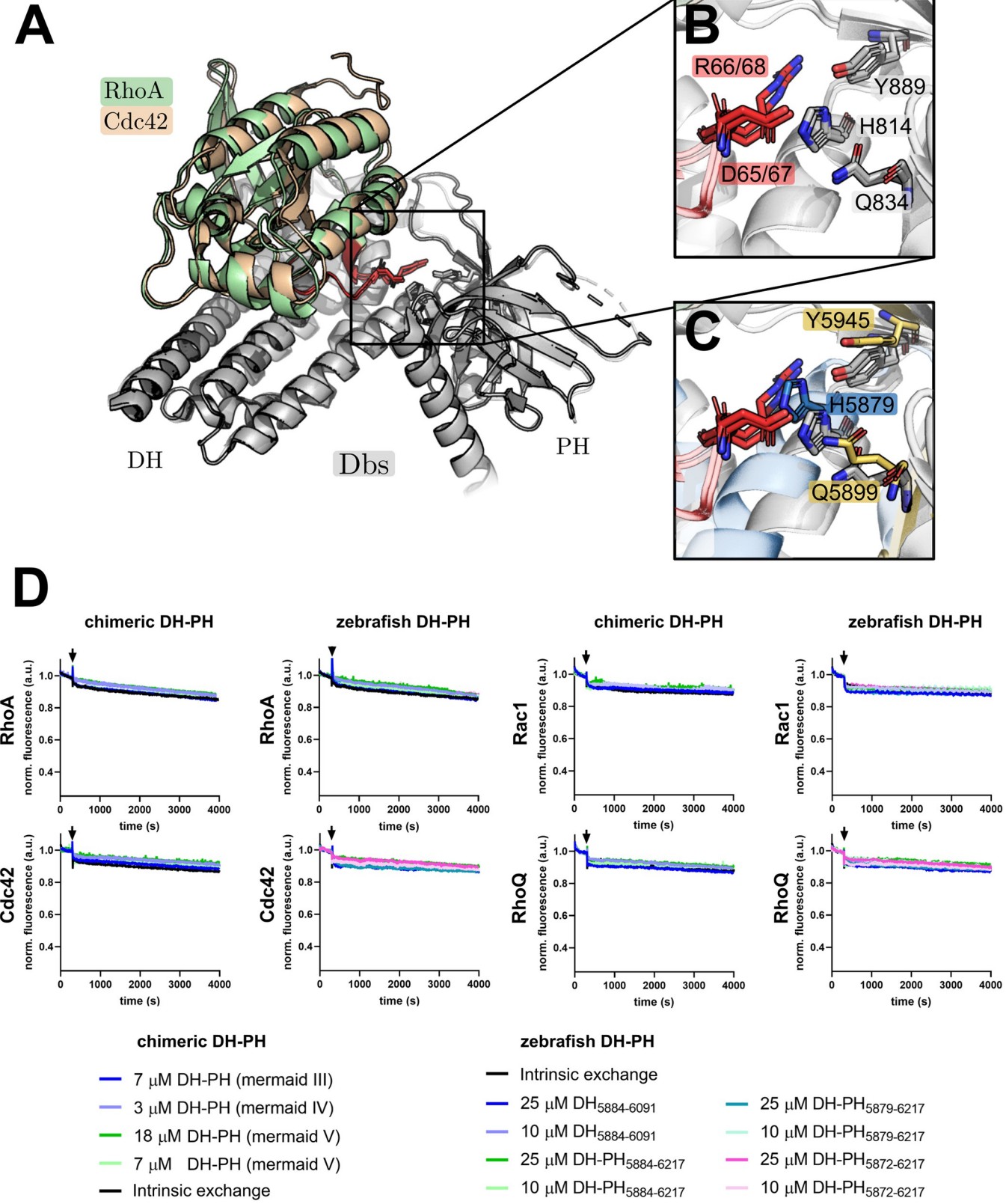

**Fig 3. Influence of the obscurin PH domain on nucleotide exchange activity.** A-C, Structural determinants of nucleotide exchange activity in Trio-subfamily GEFs. A, the aligned structures of Dbs bound to RhoA (PDB-ID: 1LB1) and Cdc42 (PDB-ID: 1KZ7) show a contact between the DH-PH domain interface and

the switch II region (red) of the respective substrate GTPase. B, a closer view of this contact shows a polar interaction between a switch II arginine and aspartate (red) and three conserved residues of the GEF (grey), two of which belong to the PH domain (Tyr889 and Gln834). C, Alignment of the predicted structure of the human obscurin DH-PH to the Dbs RhoA/Cdc42 complex shows that the corresponding obscurin residues are in a similar position as in the Dbs/substrate GTPase complexes. D, Guanosine nucleotide exchange factor activity of chimeric and zebrafish obscurin RhoGEF fragments towards RhoA, Cdc42, Rac1 and RhoQ/TC10. Black arrows indicate addition of buffer/GEF/EDTA. Data represent mean of n = 2–3 experiments. See Fig 2 for controls (experiments have been performed simultaneously in a multi-well plate reader).

To identify kinases that can phosphorylate the RhoGEF region, we used the largest human RhoGEF fragment available to us (SH3-DH) as a substrate in a commercial screen comprising a library of 245 Ser/Thr-protein kinases. Of these, 42 kinases exhibited a significant activity towards obscurin SH3-DH. Interestingly, the list of significant hits contained many CaMK-family kinases (CAMK1D, CAMK2B, CAMK2D and CAMK4) and MST-family kinases (MST1, MST2 and MST4), which also exhibited high total phosphorylation levels of obscurin SH3-DH (**S13 Fig**, panel A). A follow-up assay using the three kinases with the highest activity ratios (TBK1, CaMK4 and MST2) confirmed the activity (**S13 Fig**, panel B). While the relevance of CaMKs to cardiovascular biology is well documented, MST2 has been implicated e.g. in cardiac hypertrophic signalling in response to pressure overload [56].

To independently validate the phosphorylation of obscurin and narrow down the phosphorylation sites, we analysed phosphorylation of different GEF fragments (SH3, SH3-DH and DH) after addition of PKA, CaMK1d, CaMKII or MST2 using a phospho-protein specific staining method. In contrast to the screen, we found that only MST2 can robustly phosphorylate obscurin SH3-DH (**Fig 4A**). Interestingly, CaMKII did not phosphorylate the SH3-DH fragment but only the isolated SH3 domain, suggesting that the phosphorylation site is sterically inaccessible in the SH3-DH fragment, potentially due to an intramolecular interaction between SH3 and DH domain. None of the four tested kinases could phosphorylate the DH domain.

We further tested whether phosphatases PP1 and PP2A, both of which are crucial regulators of myofilament and SR proteins, can dephosphorylate obscurin SH3-DH following phosphorylation by MST2. We found that both PP1 and PP2A can dephosphorylate the SH3-DH fragment (**S13 Fig**, panel C).

Next, we tested whether MST2 can phosphorylate the zebrafish DH-PH fragment and a human N- and C-terminally slightly extended DH domain featuring most of the interdomain linkers including the Ser5669 site. We found that MST2 can phosphorylate the extended human DH domain, but not the zebrafish DH-PH fragment (**Fig 4B**). Finally, we directly mapped the MST2 phosphorylation site using mass-spectrometry and confirmed that it is likely Thr5798 (**S14 Fig**).

Having identified a kinase that phosphorylates the DH domain at a phosphorylation site reported in a physiological context, we tested whether this modification leads to the activation of obscurin GEF activity by repeating the GEF activity assays with obscurin that had been phosphorylated *in vitro* by MST2. We found that phosphorylation of either SH3-DH or the N-terminally extended DH domain did not lead to discernible GEF activity towards RhoA, Cdc42, Rac1 or RhoQ/TC10 (**Fig 4C**).

Neither human DH and SH3-DH (phosphorylated or unphosphorylated), nor zebrafish DH and DH-PH fragments showed any RhoGEF activity so far. Therefore, we also tested the zebrafish SH3-DH-PH domain triplet and the zebrafish DH-PH tandem both in the absence and presence of the phospholipid P(3,4)IP$_2$, which has previously been reported to bind to the obscurin PH-domain [57]. Again, no activity was detectable towards the tested GTPases RhoA and RhoQ/TC10 (**Fig 5**), suggesting that neither addition of P(3,4)IP$_2$, nor the presence of all three obscurin GEF domains is sufficient to obtain an active GEF.

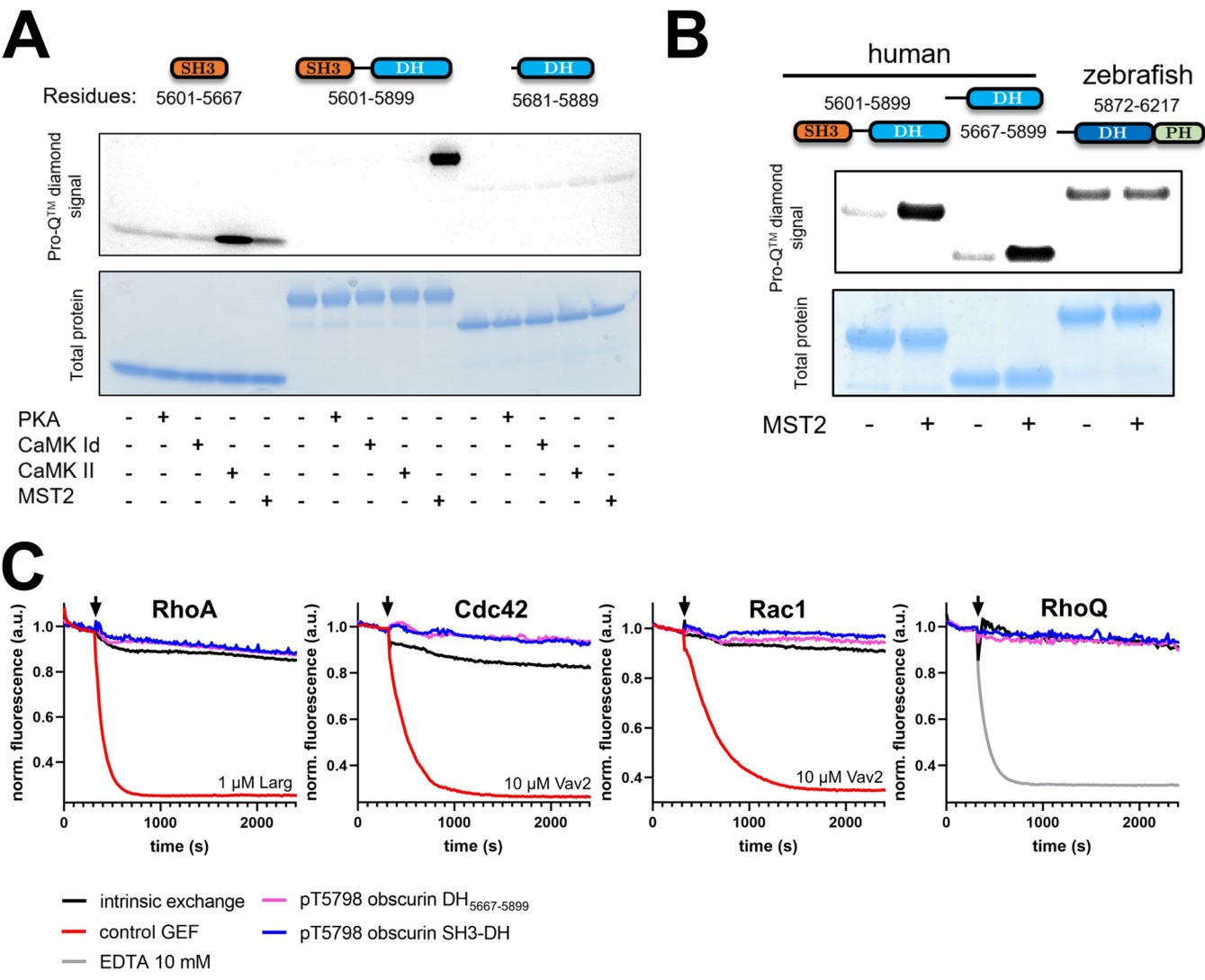

**Fig 4. Phosphorylation of obscurin RhoGEF domains.** A, phosphorylation of different human obscurin RhoGEF domain combinations by different kinases. B, MST2-phosphorylation of an N- and C-terminally extended human DH domain and a zebrafish DH-PH fragment compared to human SH3-DH. C, Guanosine nucleotide exchange factor activity of obscurin RhoGEF (10 μM) fragments phosphorylated at Thr5798 towards RhoA, Cdc42, Rac1 and RhoQ/TC10. Black arrows indicate addition of buffer/GEF/EDTA. Data represent mean of n = 2–4 experiments.

Given the lack of activity *in vitro*, we wondered whether the obscurin DH-PH tandem could change the localization of endogenous RhoA and RhoQ, which might indicate GEF activity in cells. We thus validated commercially available RhoA and RhoQ/TC10 antibodies by immunoblotting and confirmed their suitability for immunofluorescence imaging using mouse *tibialis anterior* sections (**S15 Fig**, panels A and C). Next, we overexpressed the GFP-tagged human obscurin DH-PH domains in neonatal rat cardiomyocutes using an adenoviral vector (cf. **S15 Fig**, panel B for validation of the viral constructs) and studied the localization of endogenous RhoA and RhoQ (**S15 Fig**, panel D). In uninfected neonatal rat cardiomyocytes, RhoQ/TC10 exhibited a striated localisation and RhoA a mostly diffuse localisation (although very rarely also a striated pattern). RhoQ/TC10 and RhoA localisation in cells overexpressing obscurin DH-PH did not appear any different to the cells expressing GFP alone, both in the presence and absence of the adrenergic agonist phenylephrine. Qualitatively identical results

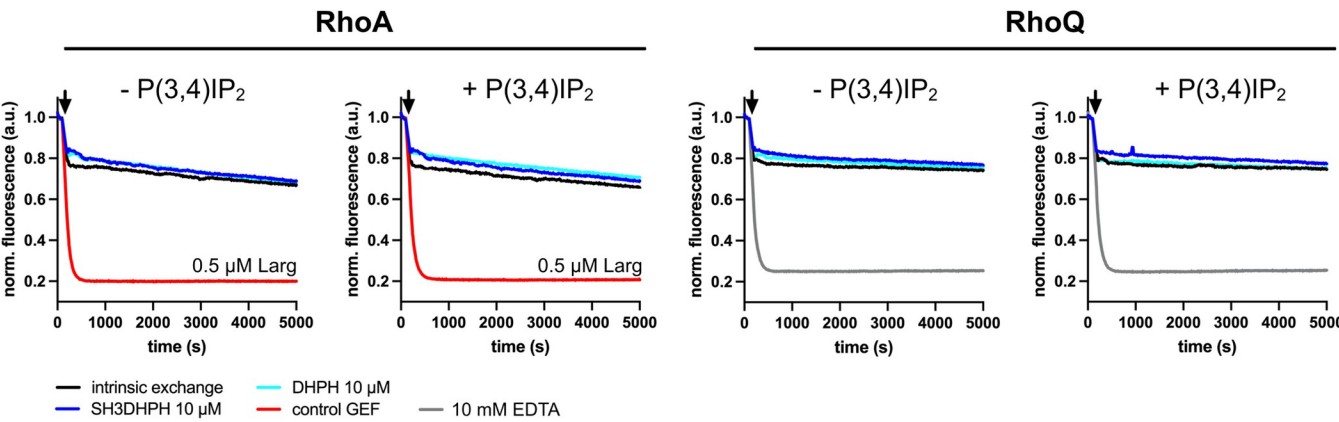

**Fig 5. Guanosine nucleotide exchange factor activity of obscurin RhoGEF fragments in absence or presence of 10 μM P(3,4)IP$_2$ towards RhoA and RhoQ/TC10.** Black arrows indicate addition of buffer/GEF/EDTA. Data represent mean of n = 3 experiments.

were obtained with adult murine cardiomyocytes (not shown). Thus, we found no evidence for activity of obscurin RhoGEF domains in living cells. However, we cannot rule out the possibility that obscurin might activate RhoA or RhoQ/TC10 without changing their localisation, or that under different culture conditions or by using alternative cell types a difference would be revealed, warranting further work to test these possibilities.

## Discussion

Although trio-subfamily GEFs are known for their reliance on the PH-domain for effective nucleotide exchange [41, 47], the complete absence of any catalytic activity of the recombinant obscurin GEF domains towards any of the 10 tested GTPases is surprising, given the previously reported activity of obscurin towards RhoA and RhoQ/TC10 using co-IP/pulldown assays [14, 15]. The inclusion of the PH domain or Thr5798 phosphorylation in obscurin did not lead to activation of its RhoGEF function either. However, when used at sufficiently high concentrations such as in our study, even inefficient DH domains typically show some degree of discernible activity towards their substrate GTPases [47].

There are multiple reasons which could account for this discrepancy. Firstly, while our experiments tested many combinations of GEF fragments and RhoGTPases, they were, due to practical limitations, not exhaustive. Thus, we might have just missed the right GEF/substrate pair which would lead to discernible GEF activity (e.g. pThr5798-SH3-DH and RhoG). However, in a systematic kinetic study of 21 DH-domain RhoGEFs, all of the tested GEFs exhibited at least some discernible residual GEF activity towards RhoA, Rac1, or Cdc42, even when these were not the main physiological substrates [44]. Since our minimal tested set of potential substrate GTPases included RhoA, Rac1, Cdc42 and RhoQ/TC10, we consider this possibility unlikely.

Another possibility is that the use of zebrafish obscurin or human/zebrafish chimeras might not work due to critical differences in the sequence and mechanism between species. Human and zebrafish RhoA exhibit a very high (>90%) sequence identity [48] and zebrafish obscurin PH residues known to be important in the GTPase/PH-domain interaction, too, are conserved as shown above. The conservation of molecular function is a feature often observed in the evolution of Ras-family GTPases. Yeast Ypt1, for instance, can be substituted by its mouse homologue without loss of function [58]. For RhoGTPases in particular, evolutionary diversification of function rather occurs via diversification of their 'regulatome' [46]. For these reasons, we

think species differences are not the most likely explanation for the lack of observed activity either.

A more likely explanation, in our view, is that the activation of obscurin requires other, potentially even multiple factors to obtain an active GEF. For instance, we do not yet know what effect phosphorylation of Ser5669 has on obscurin function or what the responsible kinase is for this site. Perhaps activation of obscurin RhoGEF function even requires both DH and PH domains plus phosphorylation at either or both Ser5669 and Thr5798 or phosphorylation-dependent recruitment of additional cellular cofactors. We conducted molecular dynamics simulations using a structural model of the SH3-DH-PH domains phosphorylaed at Ser5669, Thr5798, at both positions or at neither (**S3 Text** and **S16 Fig**). While phosphorylated structures showed no large interdomain rearrangements, nuanced differences to the unphosphorylated structure were observable for the pThr5798 and pSer5669/pThr5798 species that, interestingly, are predicted to decrease the probability of RhoA binding. Thus, phosphorylation of pThr5798 is unlikely to activate RhoGEF activity but perhaps alters other properties of obscurin such as changing the availability of binding sites for non-GTPase interaction partners. Furthermore, it is worth pointing out that we used cardiomyocytes for cell biological experiments, whereas Ford-Speelman and colleagues used *tibialis anterior* muscles [15], suggesting that such additional factors required for GEF activity might even be tissue specific. It is interesting to note in this context that the complete knockout of obscurin in mice leads to a fairly mild muscle phenotype with disrupted organisation of the sarcoplasmic reticulum and altered cellular calcium-handling [6, 59], while human loss-of-function mutations in OBSCN, abrogating the C-terminal GEF domain, display a compatible phenotype with susceptibility to severe rhabdomyolysis due to defective calcium-handling [60]. Neither phenotype offers an obvious mechanistic link to defective small GTPase signalling, other than potentially in membrane remodelling during SR-formation.

Another possibility to consider is that the obscurin RhoGEF domains might have lost its catalytic function during evolution. Enzymes which lost their catalytic function during evolution include proteins from various classes including GTPases, kinases and phosphatases [61, 62]. The N-terminal kinase of the *C. elegans* obscurin homologue unc89, for example, is predicted to be inactive due to the alteration of key residues involved in catalysis and the absence of the N-terminal kinase lobe [63]. Such enzymes are often called pseudo-enzymes and obscurin might thus be the first instance of a "pseudo-GEF". In this case, the observed effect of obscurin on RhoGTPase activity [14, 15] could be indirect. Obscurin might, for example via a non-catalytic interaction, recruit GTPases to specific membrane compartments where they could be subsequently activated by another GEF. Furthermore, pseudo-enzymes often act as scaffolding proteins and the reported interaction of the obscurin DH domain with RanBP9 [64], itself a scaffolding protein [65], might indicate the possibility of larger obscurin-associated protein complexes in the cell.

A more detailed bioinformatic analysis of the trio-subfamily DH-PH domain sequences suggests that obscurin does not cluster with other trio-subfamily members and shows the least similarity to other subfamily members (**S17 Fig**, panel A). Furthermore, obscurin exhibits several differences at multiple highly conserved and functionally important residues for the interaction with GTPases [44] (**S17 Fig**, panel B). These analyses, together with our experimental data, demonstrate that obscurin is a GEF that, if catalytically active, appears to be regulated in a more complex fashion than other RhoGEFs. Clearly, these results warrant further investigation into the activity of the obscurin DH-PH domains in living cells. A promising approach would be to use FRET-biosensors, which are available for both RhoA and RhoQ/TC10 (as well as many other RhoGTPases) to monitor the activity of RhoA or RhoQ/TC10 in response to obscurin in real-time [66, 67].

Despite the lack of observed catalytic activity *in vitro*, the availability of highly pure recombinant protein fragments paves the way for the identification and structural or biophysical characterization of further interaction partners. Moreover, the identification of multiple kinases such as CaMKs and MST-kinases as potential regulators of obscurin RhoGEF function opens new avenues to study the role of these domains in cells.

## Supporting information

**S1 Fig. Purified obscurin domains.** SDS-PAGE analysis of obscurin SH3-DH (amino acids 5601–5899) and SH3 domains (amino acids 5601–5667) after affinity purification and size-exclusion chromatography.
(PNG)

**S2 Fig. Obscurin fragment screening.** Construct design, primers used and dot-blot analysis of the eluted fraction after affinity purification of the human obscurin DH, DH-PH and PH domains.
(PNG)

**S3 Fig. Obscurin fragment screening.** Construct design, primers used and dot-blot analysis of the eluted fraction after affinity purification of the human obscurin DH, DH-PH and PH domains (continued).
(PNG)

**S4 Fig. Obscurin fragment screening.** SDS-PAGE analysis of selected fragments after affinity purification. The same samples shown in the dot-blots of S1 and S2 Figs were used for SDS-PAGE.
(PNG)

**S5 Fig. Obscurin standard scale purification.** Left, gel-filtration elution profile of DH domain fragment comprising residues 5667–5899. Middle, SDS-PAGE analysis of samples at different steps of purification process of $DH_{5667-5899}$ fragment. Right, 1D-NMR analysis of purified $DH_{5667-5899}$ fragment. NMR spectrum exhibits wide peak dispersal and peaks below 0 ppm (black arrow), indicating that the protein is folded.
(PNG)

**S6 Fig. Guanosine nucleotide exchange factor activity of obscurin RhoGEF fragments towards RhoB, RhoC, Rac2, Rac3 and RhoG.** Black arrows indicate addition of buffer/GEF/EDTA. Data represent mean of n = 2–3 experiments.
(PNG)

**S7 Fig. Pulldown experiments using obscurin $DH_{5681-5889}$ as a ligand and GST or GST-RhoA as a bait.** Addition of EDTA or alkaline phosphatase lead to a nucleotide free state of RhoA.
(PNG)

**S8 Fig. Sequence analysis of amino acids at the DH-PH domain interface.** Black arrowheads indicate important catalytic contribution (numbering refers to Dbs sequence). Conserved residues are highlighted in blue with colorintensity highlighting the degree of conservation. Top panel shows comparison of human Trio-subfamily GEFs. Bottom panel shows obscurin DH-PH domain interface across different species. The unique Uniprot-identifier for each protein is given in brackets. When the protein was not available on Uniprot (*), the NCBI-Reference sequence is given instead.
(PNG)

**S9 Fig. Chimeric obscurin RhoGEF fragments and dot-blot analysis of the eluted fraction after affinity purification of the indicated domains.** The unique Uniprot-identifier for each protein is given in brackets. When the protein was not available on Uniprot (*), the NCBI-Reference sequence is given instead.
(PNG)

**S10 Fig. Further human/zebrafish chimeras and zebrafish obscurin RhoGEF fragments.** A, construct design. The unique Uniprot-identifier for each protein is given in brackets. When the protein was not available on Uniprot (*), the NCBI-Reference sequence is given instead. B, SDS-PAGE analysis after affinity purification of selected zebrafish and chimeric obscurin Rho-GEF fragments. C, 1D-NMR analysis of purified zebrafish DH-PH5884-6217 fragment. NMR spectrum exhibits wide peak dispersal and peaks below 0 ppm (black arrow), indicating that the protein is folded.
(PNG)

**S11 Fig. Guanosine nucleotide exchange factor activity of zebrafish obscurin DH-PH$_{5884\text{-}6217}$ towards RhoB, RhoC, Rac2, Rac3 and RhoG.** Black arrows indicate addition of buffer/ GEF/EDTA. Data represent mean of n = 2–3 experiments.
(PNG)

**S12 Fig. Conservation of phosphorylation sites in the human obscurin RhoGEF region reported by Potts et al. 2017.** Numbering refers to human obscurin B sequence.
(PNG)

**S13 Fig. Top 50 results of a commercial kinase screen using [$\gamma^{33}$]-ATP and human obscurin SH3-DH as substrates.** A, (top panel) corrected absolute phosphorylation (orange) of the SH3-DH domain after exposure to a kinase next to the autophosphorylation background signal of that kinase (blue). (Bottom panel) top 50 hits of kinases sorted by the activity ratio of each experiment which considers the corrected absolute phosphorylation of the substrate relative to the phosphorylation background. A value >3 is considered a significant hit. B, validation experiment with same method of top 3 hits in screening assay at different substrate concentrations and n = 3 replicates per concentration (** $p < 0.01$, *** $p < 0.001$, students t-test vs no kinase condition). While MST2 addition resulted in strong and saturable phosphorylation, TBK1 led to much lower phosphorylation levels and CaMK4 addition led to an intermediate phosphorylation level exhibiting a biphasic behaviour with phosphorylation levels decreasing at higher substrate concentrations. Since MST2 showed the highest phosphorylation levels of obscurin SH3-DH, we focused on MST2 phosphorylation in all further experiments. C, ProQ$^{TM}$ diamond stain signal of MST2-phosphorylated obscurin SH3-DH after addition of no phosphatase (control), phosphatases PP1, PP2A or alkaline Phosphatase.
(PNG)

**S14 Fig. Identification of MST2 phosphorylation site within the human obscurin RhoGEF region via mass-spectrometric analysis of the digested phosphoprotein.** Workflow shown left. Protein sequence and identified peptides and phosphopeptides are shown on the right. Although both pSer5797 and pThr5798 peptides were identified, the precision of the identified site is often associated with an uncertainty of 1 or 2 residues. Since Thr5798 was observed to be phosphorylated *in vivo* by Potts *et al.* 2017, we concluded that the phosphorylation site is likely Thr5798.
(PNG)

**S15 Fig. Validation experiments and localisation of RhoA and RhoQ/TC10.** A, Western blot validation of RhoA and RhoQ/TC10 antibodies using recombinant GTPases shows that

antibodies specifically bind their target epitope. B, Upper: Western blot confirming expression of obscurin DH-PH. HEK293 cells transduced with adenovirus containing GFP tagged obscurin DH-PH (Ad DHPH) or GFP alone (Ad GFP) and probed with anti-HIS, anti-GFP or anti-obscurin antibodies. Lower: ponceau stain. C, Mouse skeletal muscle (tibialis anterior) stained with upper: rhoQ/TC10 (green) and z-disk titin (red) or lower: rhoA (red) and z-disk titin (green). D, rhoA and rhoQ/TC10 localisation does not change upon overexpression of GFP tagged obscurin DH-PH in neonatal rat cardiomyocytes. Upper row: untransduced cells. Middle row (GFP DH-PH): cardiomyocytes transduced with adenovirus containing GFP obscurin DH-PH. Bottom row (GFP): cardiomyocytes transduced with adenovirus containing GFP only. L: rhoA (red), z-disk titin (blue). R: rhoQ/TC10 (red), myomesin (blue).
(PNG)

**S16 Fig. Molecular dynamics simulations to assess the effect of phosphorylation on obscurin SH3DHPH.** A, Structure of Obscurin SH3DHPH domain triplet predicted using AlphaFold2. The SH3, DH and PH domains are shown in orange, blue and green, respectively. The phosphorylated residues and conserved residues required for RhoGEF activity are labelled and shown as sticks. B, plot showing mean root-mean-squared deviation (RMSD) of structures across the simulations for WT obscurin SH3DHPH and phosphorylated at either serine 5669 (pSer), threonine 5798 (pThr), or both (pSer/pThr). C, plot of the mean root mean squared fluctuation (RMSF) per residue for the three modified proteins compared to WT. D, principal component (PC) 1 and 2 heatmaps for the four molecular species. E, PC 2 and 3 heatmaps for the four molecular species. F-I, structures representing the minima (blue) and maxima (red) for PC2 (F-G) and PC3 (H-I). Arrows indicate the location of the numbered RMSF peaks in C. J, The percentage of structures in the molecular dynamic trajectories for each molecular species that do not clash with RhoA when aligned with Dbs in the Dbs-RhoA crystal structure. Representative models are shown of obscurin SH3DHPH in a binding-compatible (K) and binding-incompatible (L) conformation with RhoA (light green). Arrows indicate clashing (red) and non-clashing (green) regions.
(PNG)

**S17 Fig. Bioinformatic analysis of the obscurin RhoGEF region amino acid sequence.** A, multiple sequence alignments of the DH-PH domain sequences of all trio-subfamily members. Heatmap of the percent identity values on the left shows that obscurin is the only GEF that does not cluster (orange squares) with other subfamily members. We defined clusters as largest possible square neighborhoods along the diagonal that have at least 35% sequence identity. This is confirmed by plotting the average identity values for each GEF depicted in the right bar graph, showing that obscurin has the least average identity to other members of the trio-subfamily. B, analysis of key residues in the DH domain that have been identified to be constitutively involved in the interaction with GTPases from all Rho-subfamily members (Rho, Rac, Cdc42) in 13 DH/GTPase complex structures (Jaiswal et al. 2013). From 30 such functionally important and highly conserved residues (black arrowheads), obscurin shows significant differences at 7 of those residues (highlighted by red squares) in the regions alpha 6/7, CR3 and alpha 13 of the DH domain.
(PNG)

**S1 Raw images.**
(PDF)

**S1 Table. Primers used in this study.**
(DOCX)

**S1 Text. Info sheet KinaseFinder.**
(PDF)

**S2 Text. Technical specifications mass spectrometry.**
(DOCX)

**S3 Text. Molecular dynamics simulations.**
(DOCX)

## Acknowledgments

We thank Dr Thomas Kampourakis for access to the plate reader. We acknowledge support by CREATE [68]. The authors declare that they have no conflict of interest.

## Author Contributions

**Conceptualization:** Daniel Koch, Mathias Gautel.

**Formal analysis:** Daniel Koch, Kutti J. Vanaanen, Andrew Beavil, Martin Rees.

**Funding acquisition:** Mathias Gautel.

**Investigation:** Daniel Koch, Ay Lin Kho, Atsushi Fukuzawa, Kutti J. Vanaanen, Andrew Beavil, Martin Rees.

**Methodology:** Daniel Koch, Ay Lin Kho, Atsushi Fukuzawa, Alexander Alexandrovich, Martin Rees.

**Project administration:** Mathias Gautel.

**Resources:** Mark Pfuhl.

**Supervision:** Mark Pfuhl, Martin Rees, Mathias Gautel.

**Visualization:** Daniel Koch.

**Writing – original draft:** Daniel Koch.

**Writing – review & editing:** Ay Lin Kho, Alexander Alexandrovich, Mark Pfuhl, Martin Rees, Mathias Gautel.

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
