## [Decision Letter · Decision Letter 0]

16 Nov 2022

PONE-D-22-28822Obscurin Rho GEF domains are phosphorylated by MST-family kinases but do not

exhibit nucleotide exchange factor activity towards Rho GTPases in vitroPLOS ONE

Dear Dr. Koch,

Thank you for submitting your manuscript to PLOS ONE. After careful consideration, we feel that it has merit but does not fully meet PLOS ONE’s publication criteria as it currently stands. Therefore, we invite you to submit a revised version of the manuscript that addresses the points raised during the review process.

Your manuscript was reviewed by two knowledgeable referees in the area. As noted in the attached comments, both reviewers have felt that the manuscript is technically sound, and the data support the conclusions. In addition, the manuscript is presented in an intelligible fashion and written in standard English. Although both reviewers acknowledge the interesting results in the manuscript, they have felt that the manuscript might benefit from additional experiments, e.g. functions of phosphorylation mutants T/A or T/D in transduced/transfected muscle cells may provide further insights.

We look forward to receiving your revised manuscript.

Kind regards,

Laszlo Buday

Academic Editor

PLOS ONE

2. Please include the methodology of this manuscript as part of the main text instead of a supplementary file, as per our guidelines (https://journals.plos.org/plosone/s/submission-guidelines#loc-manuscript-organization)

“We thank Dr Thomas Kampourakis for access to the plate reader. DK was funded by a PhD studentship from the British Heart Foundation (grant [FS/17/65/33481]). This work was supported by funds from the Wellcome Trust (Collaborative Award in Sciences 201543/Z/16/Z to MG), the European Research Council under the European Union’s Horizon 2020 Programme (ERC-2019-SyG, grant no. 856118 to MR and MG) and the Medical Research Council (MR/R003106/1 to MG and ALK). Earlier work leading to the present paper was supported by the Volkswagen Foundation Research grant I/81 797 and I/78 989 to MG. MG holds the BHF Chair of Molecular Cardiology. We acknowledge support by CREATE42 . The authors declare that they have no conflict of interest.”

“DK: grant [FS/17/65/33481], British Heart Foundation, https://www.bhf.org.uk/

MG:

201543/Z/16/Z, Wellcome Trust, https://wellcome.org/

Research grant I/81 797 and I/78 989, Volkswagen Foundation, https://www.volkswagenstiftung.de/en/foundation

MG and MR:

ERC-2019-SyG grant no. 856118, european research council, https://erc.europa.eu/

MG and ALK:

MR/R003106/1, Medical research council, https://www.ukri.org/councils/mrc/

Reviewers' comments:

Reviewer's Responses to Questions

**Comments to the Author**

1. Is the manuscript technically sound, and do the data support the conclusions?

Reviewer #1: Yes

Reviewer #2: Yes

2. Has the statistical analysis been performed appropriately and rigorously? 

Reviewer #1: Yes

Reviewer #2: Yes

3. Have the authors made all data underlying the findings in their manuscript fully available?

Reviewer #1: Yes

Reviewer #2: Yes

4. Is the manuscript presented in an intelligible fashion and written in standard English?

Reviewer #1: Yes

Reviewer #2: Yes

5. Review Comments to the Author

Reviewer #1: This intriguing manuscript is an important step towards characterizing the biological functions of the tandem SH3-DH-PH domain “trio”. However, while there are extensive data that describe expression, purification and biological (in)activity of obscurin SH3-DH or DH-PH domain fragments, the authors fail to include experiments that use the full length SH3-DH-PH domain trio in experiments where its expression would have been feasible (e.g. transduction of cells). Additional experiments that are aimed at elucidating functions of phosphorylation mutants T/A or T/D in transduced/transfected muscle cells may provide further insights and clues, which should ideally be included in the manuscript.

Specific comments

Did the authors try and generate a ‘mermaid’ version containing human SH3, human DH and zebrafish PH domains to obtain soluble protein for subsequent biochemical analyses? Could it be that these domains may require their mutual presence in order to exert biological activity?

There is a report that the obscurin PH domain interacts with lipids (e.g. PIP2; Ackermann et al. JMCC 2017). Did the authors try and recapitulate this binding, and/or test if presence of lipids affects the biological RhoGEF activity towards RhoA, Cdc42, Rac1 or RhoQ/TC10?

Did the authors try and express the full SH3-DH-PH tandem in cells and test for a change to RhoA and RhoQ localization?

Figure 14 only seems to show transduction of neonatal cardiomyocytes? The manuscript states that also adult murine cardiomyocytes were tested?

There seem to some experimental results in the supplemental data that are not mentioned in the main text of the manuscript. Specifically, Figure S14 shows several validation experiments (Panels A-B) and immunofluorescence of mouse skeletal muscle (Panel C) that is not mentioned anywhere in the manuscript test.

It is a bit strange that the authors see RhoA localization in adult skeletal muscles at the Z-disc (Figure S14C). Several manuscripts describe RhoA localization at the M-band in adult tissues, including a manuscript the authors cite (Ford-Speelman et al.). In addition, RhoA localization at M-bands was lost in obscurin knockout muscles (Lange et al. MBoC. 2012), suggesting at least compartmentalisation of RhoA by obscurin. How do the authors explain this discrepancy?

If obscurin does not exert RhoGEF activity towards RhoA (etc), could the authors speculate what additional biological functions the ‘pseudo-GEF’ could perform (scaffolding, …)? An experiment that mutates any of the identified phosphorylation sites affect RhoA localization (T/D or S/D mutants vs. T/A or S/A mutants or wildtype) may bring some more insights.

The coverage of the protein using mass-spec seems to be very low (Figure S13), suggesting that the authors may miss out on phosphorylation sites in peptides that were not detected by this method. Did the authors try an alternative method (e.g. phos-tag gel analysis of in vitro-phosphorylated DH-PH fragments, or usage of a Thr5798Ala mutant) to see if all phosphorylation by MST is abrogated or if there are additional phosphorylation sites present in the fragment?

How did the authors verify that the domains are properly folded? Did the authors e.g. perform CD spectrum analysis to measure secondary structure and compare with expected 2nd structure content based on bioinformatics analyses?

Reviewer #2: The manuscript “ Obscurin Rho GEF domains are phosphorylated by MST-family kinases but do not exhibit nucleotide exchange factor activity towards Rho GTPases in vitro” by Koch et al., provides interesting new data on the still debated physiological role of the SH3-DH-PH domain of Obscurin.

Indeed,using recombinant obscurin GEF domains to test nucleotide exchange activity against representative small GTPases, they did not detect any activity by the DH-PH domains of Obscurin, suggesting that these domains are atypical and, likely subject to a complex regulation in vivo.

As pointed out by the authors, the absence of catalytic activity exhibited by the Obscurin Rho-GEF is surprising and in divergence with data from Ford-Speelman et al. and Coisy-Quivy et al., on mammalian Obscurin Rho-GEF domain, and those of Quadota et al. (JMB 2008) on the C.elegans homolog of Obscurin Rho-GEF.

Experimental approach is elegant and strongly supported by a well-studied bioinformatics analysis. Although based on almost exclusive in vitro approach, the authors present convincing data pointing to inability of the DH-PH to phosphorylate up to nine different GTPases. In addition they also verify the ability of MST2 to phosphorylate the human DH domain, paving the way to further specific studies toward the intracellular regulation of such domain.

The results are adequately discussed, underlying how these new data fit/disagree with previous evidence, and, importantly, honestly discussing the limitations that the in vitro approach may have on the general view of the biological/physiological role of the Obscurin trio domain.

Comments:

1) I have a question about data discussion: why the authors seem to ignore that the entire SH3-DH-PH domain has never been tested in their experiment? The authors themselves introduce the RhoGEF domain as a “ SH3-DH-PH domain triplet”. Indeed, the requirement of the entire domain may represent a pre-requisite for proper functioning of the RhoGEF domain. This should be discussed.

2) In light of these unexpected results, validating the in-vitro approach with in-vivo experiments becomes crucial. In figure S14 the authors tried to obtain such validation. In the system used in this work (neonatal cardiomyocytes), endogenous RhoA does not show a recognizable association with any particular region of the sarcomere as opposed to what was previously shown in skeletal muscle models. In addition, exogenous expression of the GFP-tagged DH-PH domain results in the formation of what might seem (at least judging by the reported images) as protein aggregates, possibly suggestive of protein misfolding, also not clearly associated with a particular region of the sarcomere. TC10 shows a more defined distribution, but the problem of the expression and localization of GFP-DH-PH persists. For the reasons above, I do not think that figure S14 provides valuable information as it is now. I would suggest using a different system (e.g. derived from skeletal muscle) to judge the potential rearrangement of Rho and TC10 in the presence of the Rho-GEF domain and possibly testing different GFP-tagged Rho-GEFs construct to evaluate and improve the quality of their exogenous expression and localization.

3) The phosphorylation by CAMKII of the SH3 domain but not of the SH3-DH domain is possibly indicative of intramolecular interactions between the SH3 domain and the downhill DH. Such interactions might be important for the catalytic activity of the DH domain. The authors created different constructs, including a couple (Mermaid II and a zebrafish construct) containing the entire SH3-DH-PH fragment. Those constructs, though, were not tested for their catalytic activity. Is there a reason? Would it be possible to see data about the catalytic activity of these fragments on different GTPases to rule out a possible role of the SH3 domain in the catalytic activity of these isolated fragments?

4) In the aligned structures reported in figure 1, it appears that a noticeable difference between Dbs and Obscurin is the presence of Histidine 5879 in very close proximity to the switch region of RhoA and Cdc42. Have the authors considered if such a difference could lead to steric hindrance for the interactions with RhoA and/or cause reduced flexibility of the switch domain with potential consequences on the catalytic activity?

minor concerns:

1) Reference 1 and 10 appear a bit dated compared to the concept expressed.

2) Figure 1D is not acknowledged

3) page7, line216: the authors introduce the experiments on RhoA and RhoQ localization as also performed on adult murine cardiomyocytes. There’s no trace of such cells throughout the manuscript.

4) Why the zebrafish DH-PH domain was used in MST2 phosphorylation assay (Fig.4B), if the Thr5798 residue is not conserved in that species, and the Ser5669 seem to be not included in the zebrafish DH-PH construct (5872-6217, according to figure 4B) ? Was it used as experimental control?

5) The authors have tested 21 DH fragments and they choose two of them. Can the authors explain the reason for this choice?

6) The authors find no nucleotide exchange activity of any of the DH or DH-PH fragments. Did the authors test higher fragment concentrations?

7) RhoQ/TC10 was found to be activated by an obscurin fragment covering the C-terminal region from 5679 to 6620. Did the authors test a similar fragment as a positive control for their nucleotide exchange activity and binding experiments?

Minor issues

The fonts used to label some figures are small and hard to read. This issue is especially noticeable in Fig1 B and Fig 3D. Please use bigger fonts.

Figure 2: The scale of the Y-axis in Rac1 + Control Graph is slightly different from all the others. It would be ideal to use the same scale for all graphs.

The location of Ser5669 is described ambiguously. In line 203, it is said to be part of the interdomain linker between SH3 and DH, while in figure S15 it is represented as part of the SH3 domain. Please correct the ambiguity.

6. PLOS authors have the option to publish the peer review history of their article (what does this mean?). If published, this will include your full peer review and any attached files.

Reviewer #1: No

Reviewer #2: No

---

## [Author Response · Author response to Decision Letter 0]

28 Mar 2023

Please see the attached "Response to reviewers" further below.

---

## [Editor Report · Decision Letter 1]

3 Apr 2023

Obscurin Rho GEF domains are phosphorylated by MST-family kinases but do not

exhibit nucleotide exchange factor activity towards Rho GTPases in vitro

PONE-D-22-28822R1

Dear Dr. Koch,

We’re pleased to inform you that your manuscript has been judged scientifically suitable for publication and will be formally accepted for publication once it meets all outstanding technical requirements.

Kind regards,

Laszlo Buday

Academic Editor

PLOS ONE
---

## [Editor Report · Acceptance letter]

12 Apr 2023

PONE-D-22-28822R1 

Obscurin Rho GEF domains are phosphorylated by MST-family kinases but do not exhibit nucleotide exchange factor activity towards Rho GTPases *in vitro*

Dear Dr. Koch:

I'm pleased to inform you that your manuscript has been deemed suitable for publication in PLOS ONE. Congratulations! Your manuscript is now with our production department. 

Kind regards, 

on behalf of

Professor Laszlo Buday 

Academic Editor

PLOS ONE